# “Did You Bring It Home with You?” A Qualitative Investigation of the Impacts of the COVID-19 Pandemic on Victorian Frontline Healthcare Workers and Their Families

**DOI:** 10.3390/ijerph19084897

**Published:** 2022-04-18

**Authors:** Jade Sheen, Elizabeth M. Clancy, Julie Considine, Alison Dwyer, Phillip Tchernegovski, Anna Aridas, Brian En Chyi Lee, Andrea Reupert, Leanne Boyd

**Affiliations:** 1School of Psychology, Deakin University, Geelong 3200, Australia; elizabeth.clancy@deakin.edu.au (E.M.C.); anna.aridas@deakin.edu.au (A.A.); brian.lee@deakin.edu.au (B.E.C.L.); 2Centre for Quality and Patient Safety, Institute for Health Transformation, School of Nursing and Midwifery, Deakin University, Geelong 3200, Australia; julie.considine@deakin.edu.au; 3Eastern Health Melbourne, Box Hill 3128, Australia; alison.dwyer@easternhealth.org.au (A.D.); leanne.boyd@easternhealth.org.au (L.B.); 4School of Educational Psychology and Counselling, Monash University, Clayton 3800, Australia; phillip.tchernegovski@monash.edu (P.T.); andrea.reupert@monash.edu (A.R.)

**Keywords:** frontline, healthcare worker, COVID-19, family functioning, pandemic

## Abstract

Concerns regarding the physical and mental health impacts of frontline healthcare roles during the COVID-19 pandemic have been well documented, but the impacts on family functioning remain unclear. This study provides a unique contribution to the literature by considering the impacts of the COVID-19 pandemic on frontline healthcare workers and their families. Thirty-nine frontline healthcare workers from Victoria, Australia, who were parents to at least one child under 18 were interviewed. Data were analysed using reflexive thematic analysis. Five superordinate and 14 subordinate themes were identified. Themes included more family time during lockdowns, but at a cost; changes in family responsibilities and routines; managing increased demands; healthcare workers hypervigilance and fear of bringing COVID-19 home to their family members; ways in which families worked to “get through it”. While efforts have been made by many healthcare organisations to support their workers during this challenging time, the changes in family functioning observed by participants suggest that more could be done for this vulnerable cohort, particularly with respect to family support.

## 1. Introduction

Healthcare workers play a crucial role in supporting the health and wellbeing of the community, especially in times of crisis. The COVID-19 pandemic has placed great demand on frontline healthcare workers (FHWs), putting their own mental health and wellbeing at risk [1,2,3]. Based on a meta-analysis of 25 studies from across the world, Varghese et al. [4] found that nurses were at higher risk of poor mental health outcomes during the pandemic, with increases in anxiety, stress, depression, PTSD and insomnia noted. Pre-COVID-19 research also found that when FHWs are exposed to infectious diseases, they struggle with isolation, nervousness, insomnia, exhaustion and other physical and mental health disorders [5,6]. Such research suggests that the mental health and wellbeing of frontline healthcare workers is a priority, particularly given the essential nature of their roles.

FHWs have experienced negative effects of inadequate organisational management throughout the pandemic, including overwork, high emotional demands and reduced rewards [3,5,7,8,9,10,11]. These stressors occurred at a time when physical activity and meditation, two techniques known to reduce stress, decreased amongst some healthcare workers [12], potentially impacting their mental health. In many countries, lockdowns have also been used to reduce community transmission of COVID-19, leading to decreased opportunities to engage in wellbeing-related activities and also to decreased social participation, even when FHWs are not at work [13]. In this context, the term lockdown refers to “a series of government mandated restrictions on the movements of most community members, in an effort to reduce the spread of COVID-19, typically occurring when the spread of the disease becomes so widespread that more precise interventions are rendered less effective” [13] (pp. 1–2).

The physical risks of frontline work add an additional layer of stress for FHWs, potentially compounding existing stressors. Shah et al. [14], for example, showed a threefold and twofold increase in admission for COVID-19 amongst FHWs and those who live with them. Unlike other occupations, frontline healthcare workers cannot protect themselves in lockdown by working from home. While distancing measures can be observed in some clinical contexts, this is not always possible in emergency situations [15]. Thus, FHWs have been conscious of their own risk of infection through the pandemic but also carry a weight of responsibility for their families, other patients and the community [5].

Outside their work roles, FHWs have societal roles such as parents, spouses and offspring [16]. As adversities and hardships are experienced not only by the individual but by their entire family [17], research must consider the impacts of the pandemic on both FHWs and their family members. In particular, researchers should seek to understand the impact of the pandemic on frontline families with young children, as a number of studies have highlighted the impacts of parental mental illness and stress on children [18,19,20,21].

There is currently a dearth of literature pertaining directly to the impacts of the pandemic on frontline families. At the time of writing, one study was identified that directly considered the impacts of the pandemic on FHWs families. Chandler-Jeanville et al. [1] interviewed 49 French nurses and their family members regarding their work during the COVID-19 pandemic. While differing views were expressed by participants, most commented on the risks of their work to their physical health and wellbeing. Positively, the pandemic was also seen to have focused attention on nurses already experiencing? tough working conditions. These authors suggested that nurses and family members became collateral damage in the pandemic, as both groups were adversely impacted by the nurses’ frontline roles.

Studies of FHWs also offer some indirect insights into the impacts of the pandemic on family units. These studies suggest that frontline families have experienced isolation and separation during the pandemic, with reports of stigma and aggression directed at the healthcare workforce [22,23], isolation from family members when quarantined due to exposure at work [24], and separations caused by increased workloads and distancing due to fears of infection [7]. Fear has also been a central characteristic of frontline work, particularly for those with families. This fear extends to FHWs’ own health and risk as well as that of their family [1,5,7]. Frequent adaptions have been required to accommodate rapidly changing policies and procedures for managing infectious patients, alongside adjustments to family routines [7]. In many cases these changes created a significant domestic burden, one that was particularly felt in female FHWs and those with dependent children [25]. Protectively, living with others has been associated with lower odds of returning a positive screen in depression and anxiety inventories amongst healthcare workers and provides an important source of social support [26,27].

Currently, no studies have investigated the impact of the pandemic on family functioning within frontline families. In this context, family functioning would include an understanding of family roles, routines and rules as well as challenges and strengths. Understanding family functioning is critical, as it has the power to help or hinder an individual’s attempt to process and cope with challenging life events, such as working through a pandemic [28]. Understanding what changes, if any, occur in family functioning through this period will allow clinicians to identity and target vulnerabilities in addition to enhancing existing resources and strengths.

### 1.1. Aims

This study aimed to explore the impact of the COVID-19 pandemic and associated lockdowns on FHWs and their families. The key research questions for this study were:What changes, if any, are observed in the functioning of frontline families during the COVID-19 pandemic and associated lockdowns?What impact, if any, does frontline work status have on healthcare workers and their families?How are frontline families supporting one another during the COVID-19 pandemic?

These questions are addressed from the perspectives of 39 frontline healthcare workers living and working in Victoria, Australia. In undertaking this research, several definitions of family were considered. As lockdowns limited access to those outside of the immediate household, definitions with a focus of household relationships were favoured. The following definition was adopted and considered during recruitment: “*a group of one or more parents and their children living together as a unit*” [29].

### 1.2. Australian COVID-19 Context

To contextualise the objectives and findings of this study, a brief overview of the Australian COVID-19 context is warranted. Following Australia’s first reported COVID-19 case in March 2020, strict social distancing and isolation guidelines were employed to manage the progression of the disease. International borders were quickly closed, and mandatory quarantine was introduced for returned travellers. Strict stay-at-home restrictions were implemented nationally from late March to early June 2020. Mandates across the country included social distancing and the use of protective equipment such as hand sanitiser and masks. Figure 1 provides an overview of key milestones in Australia’s response to the COVID-19 pandemic, including the additional social and movement restrictions imposed in the state of Victoria, where this data was collected. The data from this paper were gathered from October 2020 to February 2021. At this time, COVID-19 case numbers in Australia remained low per capita compared to most other countries in the world [30]. As a result, few frontline workers had actually contracted COVID-19. Lockdowns were still heavily utilised as a means of controlling outbreaks, particularly in Victoria, and thus were included in the research questions.

## 2. Materials and Methods

### 2.1. Participants and Recruitment

Study participants were 39 frontline healthcare workers employed in Emergency Departments, Intensive Care Units, COVID-19 wards and Hospital in the Home teams (providing in-home medical support for COVID-19-positive patients) working in Victorian metropolitan and regional hospitals, Australia. As existing research suggests that family units with dependent children have been vulnerable during the pandemic [13], this study recruited frontline healthcare workers with at least one dependent child under 18 years of age living in the home at the time of the study.

Recruitment advertisements were posted to online social media platforms (e.g., Facebook, Instagram, LinkedIn) and through healthcare networks. Interested parties then emailed Author #2 for further information and consent forms. Forty-one participants returned consent forms agreeing to participate; however, due to scheduling conflicts, 39 booked a time to complete the interview. Participant details are presented in Table 1. Participant ages ranged from 29 to 57 years (M = 41.6 years; SD = 7.1 years). Participants had 1–4 children (M = 2.1), ranging from 9 weeks to 21 years of age (these families had other dependent minors in their care). All participants had experienced exposure to confirmed or suspected COVID-19 cases within their working roles.

Attempts were also made to recruit partners and children of participants to add breadth to the results. An invitation was extended at the conclusion of the interview with participants, and separate invitations were also emailed after the interview. One partner (also a frontline healthcare worker) and one child consented to participate. Due to poor recruitment, their results were not included in this study.

### 2.2. Procedure

Ethical approval for this study was provided by the Deakin University Ethics Committees (HEAG-H 70-2020). Participants completed one, or in the case of seven participants who needed to cease their first interview early, two, semi-structured interviews. Interviews ranged from 20 to 146 min, with an average of 67 min. The interview schedule was developed by the researchers and based on the research questions and previous research regarding the impacts of disasters on family functioning and healthcare challenges during the pandemic. The McMaster Family Assessment Device (FAD [31]), while not administered directly, was also considered when developing questions pertaining to family functioning.

The final interview schedule included 15 questions, with prompts, examining participants perceptions of changes in the household: family roles, routines and rules; parenting practices; communication and relationships; the impact(s) of their healthcare roles; strengths, challenges and tensions. Participants were asked to reflect on changes “*since the beginning of the COVID-19 pandemic*” or “*during periods of lockdown*”. As an illustration, participants were asked “*what changes, if any, occurred in your family routine during lockdown?*”.

Interviews were completed between October 2020 and February 2021 by authors E.M.C, B.E.C.L and A.A. All interviews were completed via Zoom video conferencing software [32]. With participants’ permission, interviews were audio-recorded, transcribed, verified by the researchers, and sent to participants for approval. Participants were given two weeks to provide feedback on their transcript before the data were reviewed to remove any identifying details and included in the group analysis. Four participants from the total sample requested edits to their interview transcript. Three of the four removed information they felt could be identifying, and two of the four elaborated on their interview responses.

### 2.3. Data Analysis Procedure

This study applied a reflexive thematic analysis [33]. Within this approach, themes are conceptualised as meaning-based patterns evident in both explicit and conceptual ways. As lived experience of an unusual event was central, thematic analysis was completed in an idiographic manner to reduce the loss of individual experiences and perspectives. This was achieved through thorough analysis of each interview transcript before comparing across the sample. While the interview was influenced by family functioning domains, the analysis was inductive and data driven.

Authors J.S., E.M.C, P.T., A.A., B.E.C.L, and A.R. were each randomly allocated 4–8 transcripts to review. Initially, the authors undertook a preliminary analysis of each transcript, which involved reviewing each transcript, identifying quotes which held meaning for the participant, the researcher and the research questions. They then coded transcripts to identify and label meaning at a micro level. Codes were clustered to identify key concepts within the transcript. Comparisons and contrasts of these concepts were then made across transcripts through recursive and reflective discussion. Recurring concepts were identified and developed into provisional super- and subordinate themes. Authors then reviewed their allocated transcripts against the themes to ensure that they accurately represented each transcript, highlighting relevant quotes as well as contrary cases. During a second collaborative discussion, the themes were reviewed and finalised.

In creating a narrative for the results, the authors tabulated the themes and for each participant, reflected on the presence of related data or conversely, refuting data to ensure that individual perspectives were not lost. The lead author then drew from this resource as well as reviewing original transcripts to retain authenticity. The transcripts were then rechecked to examine for any specific pattern in the themes, or elsewhere, based on gender, age of their children and other demographics.

### 2.4. Reflexivity

J.S. and E.M.C. led the data collection and analytical processes and manuscript preparation. At the time of writing, the first author was a Clinical Psychologist and Associate Professor while the second author was a Psychologist and Research Fellow. Both were parents residing in Victoria, Australia, during this period, with lived experience of parenting during the pandemic and simultaneously working with families and parents as psychologists and researchers. The research team brought expertise in research methods, particularly qualitative research, experience from practice and as FHWs, in hospital and clinical settings. These experiences shaped the paradigm and viewpoint of the research team. To minimize researcher bias, maintain data integrity and analytical rigor, specific steps were taken by the authors. These included maintaining journals throughout the research, engaging in self-reflection and robust, regular discussion within the data analysis team, and including a mixed representation of genders, parents and non-parents in the data analysis team. The transcripts were also regularly reviewed to ensure that the themes generated aligned with the data provided by participants, as described in the data analysis procedure above.

## 3. Results

Through reflexive thematic analysis, five superordinate themes were identified with eleven additional subordinate themes. These themes are reflected in Table 2. Within the body of the results, quotes are italicized and each participant’s age and gender are bracketed after quotes (gender, age).

### 3.1. Time Together, but at a Cost

This theme relates globally to shifts in relational connections that were particularly observed during periods of lockdown. Participants described spending more time with family members in the household due to lockdown restrictions, which led to shifts in family connections. Lockdowns also had a relational cost, with many noting the loss of personal space and social connection.

#### 3.1.1. Time Together

Due to lockdown and the closure of many activities, families spent additional time together. Some participants enjoyed this, with Participant 8 (F, aged 38 years) noting, “*we’re all a little bit closer because we’ve spent so much time together*”, while Participant 34 (F, 39) reported that during lockdown, “*the [family] relationship is better*”.

There appeared to be some gender-based observations of these changes. As an example of the wider trend, one mother made the following observations about her (male) partner, “*We recognised how much time that he [partner] was missing out on… it was just lovely having him around and everybody really was grateful that he could work from home*” (F, 39). Another noted “*My daughter and I have always been close, but I think her relationship with her dad is a lot closer now, as well*” (F, 49).

The cessation of activities outside the family meant “*…we’ve been able to spend more time as a family because we’re not rushing off separately*” (F, 44). Opportunities for connection were found through more shared mealtimes: “*I think mealtimes, eating together … has been a good bond for us, to bring us together…to sit down and talk about what we did in our day*” (F, 54).

Not all parents observed a positive shift towards shared time, especially those with adolescent children; e.g., “*I think there’s more isolation within the household [during lockdown]. The [adolescent] kids probably spend a bit more time in their own space and bedrooms.*” (F, 56).

#### 3.1.2. Impacts on Relationships

Many participants observed positive changes within the marital dyad as a result of lockdown restrictions and increased time together; for example, “*If anything, it’s probably strengthened us in the long term because we’ve had to talk about the way we have been processing things and the feelings we have …*” (F, 33). When commenting on her relationship another participant noted, “*We did consciously try to … improve everything about our relationship because it was a nice time to do it and it was something we could work on…*” (F, 44).

For some families, child–parent connections also appeared to improve, with a representative comment being, “*I’ve probably become a lot more reflective and in tune with where they’re at [the children], because I’m not running around like a chook without a head…I’m more present*” (F, 44).

In some families, a strengthening of sibling bonds was observed, and where siblings were “*… stuck in the house together at times, it did improve their relationships*” (F, 56). Participant 33 (F, 43) noted of her children, “*I feel like they’re getting along better and understanding each other*”. Conversely, some parents reported more tension and arguments between siblings: “*My son tried to take on some of the home schooling or parenting… that created a lot of angst between the [children], so there was a little bit more than their usual fighting*” (F, 46).

#### 3.1.3. Loss of Personal Space

Family time within the household came at the expense of parents’ personal space: “*I have missed just having some time to myself and just being able to not have to talk to someone or not have to be the mum or be the wife*” (F, 42). This presented challenges; for example, “*Having the kids at home all the time, … not having a break, not having any sort of outlet…You started to feel like… the house was closing in*” (F, 35) and “*I found that she [partner] was starting to get quite stir crazy*” (M, 46). This lack of personal space was particularly challenging for the majority of participants who had increased demands at work. For example, “*There was just no rest, it was either full on at work or it was full on at home*” (F, 39).

The pressures of close contact were even more significant for the minority of participants who found themselves quarantining with their family following an exposure; for example, “*as a family… [we] spent 14 days isolating … That was a bit challenging because we were crunched together … in a hotel room*” (M, 46).

Some described actively searching for time alone, e.g., “*We needed to have time apart because being in each other’s pockets wasn’t always good… so … [I’d say] ‘I’m going to go for a walk, but I’m going to go for a walk by myself*’” (F, 31), with another indicating that “*I found sometimes the only way to get some time to myself was I’d just go to bed and I could just lie in bed for a minute on my own*.” (F, 38).

#### 3.1.4. Loss of Social Connection

Participants described missing out on normal social interactions: “*My biggest challenge was not being able to see…my extended family*” (F, 49) and “*Probably the biggest thing was…not having any support. I’ve got my mum and friends and things and I’ve got a really good … supportive network…I didn’t have any of that [during lockdown] which was really hard*” (F, 40).

Concerns were also noted for their children: “*My daughter … really missed her social group and her friends…I think she’s been… more stressed and a bit less willing to participate and contribute to the family*” (M, 46). From another participant, “*My nine year old …could jump [online] with his friends…whereas my six year old … didn’t have much contact with people during [lockdown]*” (F, 40).

### 3.2. A Time of Change

The pandemic was typically identified as a significant time of change in households, where routines, both individually and within the family unit, were modified to comply with lockdown restrictions and work demands. Responsibilities for household tasks and roles also shifted.

#### 3.2.1. Changes in Family Routines and Roles

For parents of school-aged children, most parents noted, “*Our whole routine has changed*” (F, 29), which for some “*posed some challenges…readjusting our normal*” (F, 41). Lockdowns and home schooling were often identified as the primary reason for changes. For many, creating a sense of structure within the day was an important part of managing changes in routine.
*We tried to schedule as much as we could… As soon as I got stuff from the school I’d be putting it in my calendar and then I’d be copying that onto a whiteboard so…my kids could see… that helped I think*.(F, 39)

Although positive, when restrictions were lifted and children returned to school, another set of changes and adaptations to routines for FHWs was required.*The week I was supposed to do a 7 day stretch in the unit, [child] went back to school so then I had to try and swap out of the weekend… you just couldn’t predict what was going to happen in the future … that I think was a little bit of a challenge just trying to think about how to do a juggle with a child who was going to have his routine disrupted again*.(F, 44)

#### 3.2.2. Changes in Parents’ Routines

Personal activities dropped off for some, particularly in the face of high demands and transmission fears. For example,*Because I was always working my husband couldn’t get out for that run just to release his tensions*…(F, 33)



*I probably wouldn’t got back to [Tai Chi] class or if I do, I will probably stand the furtherest away… I’m a front liner, I do not want to expose [older participants].*
(F, 45)


Sleep routines were disrupted for some: “*I haven’t been sleeping as much because of, because of everything that’s been going on*” (M, 38). A few participants also reported “*Alcohol increased more*” (F, 44). For example, “*There was probably 6 months of that time where I didn’t have an alcohol free day… It was rough during that period of time*” (M, 38).

#### 3.2.3. Changes in Responsibilities

Female participants with male partners particularly observed their partners assuming additional responsibilities during lockdown, with resulting implications for family dynamics. For example, “*he has now taken over as the lead of the household…*” The same participant continued by pointing out, “*it’s very tricky trying to change that relationship balance, particularly because you know if you have always been in control. It’s hard to let go of that control and let somebody do something different*” (F, 29).

In other families,*He’s (husband) actually done meals 3 days a week, school pick-up and drop-off every day of the week…The boys have done chores: vacuuming and mopping…And so it’s completely turned on its head… This is the silver lining.*(F, 42)



*All of a sudden he [husband] was home…obviously it was hard for me to let go of controlling everything at home as well as trying to work and letting him work it out and do it his way; so there was a bit of a teething period.*
(F, 38)


Due to their health roles, an additional responsibility observed by some participants was providing advice and support to extended family about health; for example, “f*or all of my extended family … I’m providing healthcare advice, facilitation…*” (F, 46).

However, not all participants felt qualified to provide this type of support.*Being an ICU nurse, being a healthcare worker, everyone asks you oh, what should I be doing?… I remember actually saying to my mum at one point I don’t know, I haven’t worked in a pandemic before … this is all new to me… I’ve never had to do any of this stuff either.*(F, 31)

### 3.3. Increased Demands

This theme relates to participants observations of increased demands on their time and emotional resources at work and at home. Participants described an increase in demand occurring almost simultaneously in their work and family lives, which appeared to drain their resources.

At work, participants described working more hours because they felt “*obligated to pick up extra shifts*” (F, 49) due to the high demands resulting from COVID-19. For many, this change occurred alongside an increase in the emotional complexity of their work, e.g., “*We had 7 patients die in the first week. Normally in our rehab ward we’d have 7 patients die in a whole 12 months*” (F, 56), and changes in the care allowed for patients’ family members, “*We weren’t allowing any visitors into the ICU… our patients …are critically ill or dying and not being able to let their families in was very stressful*” (F, 39).

As a result, some described being emotionally spent as work demands were “*quite draining*” (F, 39), where “*You feel like you’re kind of failing the people that you’re going to be caring for at the hospital because you’re getting there and you already feel so depleted*” (F, 39). From the perspective of Participant 36, “*As healthcare workers you just push through everything … but I just (kept) thinking…why do I have to do all of this now, I’m struggling*… (F, 39).

While most frontline work occurred in situ, participants also reported a reduction in the ability to switch off after work, with increased checking of emails from home. This was particularly evident in participants with managerial-level roles, as typified in the observation, “*I don’t actually switch off—in the evenings and over weekends—there’s always email checking…because especially in the early days of COVID… it was so dynamic*” (M, 38).

Some reported that additional work demands and lack of personal space impacted on their patience or tolerance with family: “*Sometimes I’d be a bit short and a bit more stressed…The kids probably are a little bit more clingy… once I’m home they want to be close all the time, it’s just they miss Mum…*” (F, 38).

In some cases, participants were under financial pressure to increase hours to compensate for other family members having lost work: “*…My husband was out of work, so I went back to work four days a week and he stayed home… it was a big change… I was essentially the bread winner*” (F, 38).

Balancing additional work demands with concurrent changes at home was particularly challenging:
*[Lockdown] ended up being quite a crazy few months juggling hospital shifts which I absolutely had to go to and trying to support my kids through home learning… I was [a] part time nurse, part time teacher… trying to keep my head above water…*(F, 39)

This issue was particularly salient for mothers and families with primary-school-aged children, who required assistance with their schooling:
*The kids thought… Mum’s home, she can help me with my school work but then it sort of put several hours into the school day then I’d go and do a full shift at the hospital so by the time I’d finish my workday late in the evening I was absolutely exhausted…*(F, 39)

Seeing lockdown restrictions and the additional demands as a finite experience helped some to manage the demands at home: “*Because we knew there was an end to it, we just… had to bunker down and go, okay, well, we just have to get through this*” (F, 40).

### 3.4. Hypervigilance and Risk of Contagion

As FHWs, participants were well aware of the potential risks associated with their roles. Fear and anxiety were expressed, particularly in relation to potential impacts on family members. Participants outlined the procedures they undertook to minimise risk, potentially as a means of controlling their anxiety.

#### 3.4.1. Risk Consciousness

Most participants described being aware of contagion risk, with one illustrative comment being, “*You did go home just feeling a little bit dirty… you were you know potentially bringing stuff home…because of the job you’ve been doing*” (F, 39). Due to their work, the risk of contagion was also noted in family members, e.g., “*Every time I came home, my husband would go—did you bring it home with you. Did you catch it*?” (F, 38).

Risk consciousness was more significant for staff from Emergency Departments, those with pre-existing vulnerabilities and/or those who had vulnerable family members, e.g., “*If I got sick with COVID, my ability to recover is impaired, and there’s a likelihood that it would be quite severe …*” (F, 41). The sense of concern extended to fears around fatal outcomes for their family, especially children, with one participant noting, “*I can remember feeling quite concerned and stressed about the potential for me to be bringing it home, especially to my young kids*” *(F, 35),* with another stating, “*It was definitely my own safety, my family’s safety… I was genuinely concerned that I might get something and die…*” (F, 31). Another stated, “*…[It’s] the first time I’ve ever felt scared going to work*” (M, 34).

Fears were also held by partners for FHWs, with a number of participants noting contingency planning in the event of illness, for example, “*My partner had me rewrite my will*” (F, 46). With respect to extended family, a small number of FHWs experienced stigma due to concerns regarding their frontline status.*His parents were happy to see … his brother’s kids during that time, but weren’t happy to see ours, which I didn’t want them to anyway but there was a bit of double sort of standards going on*.(F, 35)

#### 3.4.2. Managing and Mitigating Potential Risks at Home

Participants had active plans for how they would manage if they did contract COVID-19. For example, “*I was going to isolate in the back bedroom… I had my own contingency*” (M, 34), with another participant noting “*I was pretty confident that I would get sick, and I didn’t want to get the kids sick…we restructured the house into two parts that I could retreat to if I needed*” (F, 49).

Participants had considered various scenarios, such as “*if I were to bring the disease home, and suddenly my partner and I are sick, and there’s nowhere to transfer the kids*” (F, 46), and what they would do as a family to manage. Parents with older children held conversations with children about these possibilities:*I prepared them… I told them that they didn’t need to be worried if mummy couldn’t see them for a week because she’d been exposed to it, that everything would be fine, and they should expect that there might be calls or moments like that.* (F, 46)

For some, anxiety about being a transmission vector was reflected in reduced displays of affection towards their family, with one participant noting, “*I certainly kept my distance from my mum which I’ve never had to think about doing before*” (F, 39), while another participant noted, “*If [the children] wanted a hug when you came in, I said* ‘*Do not come near me*’” (M, 49). Others engaged in lengthy hygiene procedures when they returned to their homes to keep their family members safe, “*I have been taking all precautions like getting changed…when I come home there’s not going to be any touching, I’m going straight to … the spare shower…*” (F, 45).

### 3.5. Getting through It

Participants reflected on the different factors that impacted their wellbeing and mental health during lockdowns as well as the mental health and wellbeing of family members. Coping appeared to fluctuate rather than being a steady state amongst participants, with a reduction in expectations noted as one coping strategy to manage.

#### 3.5.1. Wellbeing and Mental Health

Some participants reported the increased burden of balancing work and parenting, such that their own “*mental health has been affected*” (F, 35). Participants also reported concern for their children’s wellbeing and mental health and the impacts of lockdown.*I always worry about their wellbeing and their happiness… [but] that was actually probably even more heightened…I really did worry if they were going to make it through this socially and mentally, more than anything.* (F, 40)

There were specific mental health impacts for those experiencing additional developmental change: “…*starting Year 7 again they’ve had increased anxiety about that…same as the anxiety that she had in first day…but doing it all over again*.” (M, 49)

#### 3.5.2. Coping

Participants’ ability to cope with the demands of lockdown fluctuated.*In the first lockdown, I felt like I coped with that better. … because it was so—it was such a new kind of concept … we were all in it together and I felt like I was doing okay …. But this second lockdown in particular I know that … I was less patient, I was feeling tense, I was getting—I was feeling quite stressed out …*(F, 35)

To cope, many participants tempered their expectations of themselves and their parenting.*I put way too much stress on myself, especially…the first time round and the start of the second [lockdown]…once we were in the second part of it, I just had to give up a bit really and do what I could because it was just impossible to do it all.*(F, 40)



*During the lockdown part, there wasn’t a lot of discipline. We were just all trying to just get through it the best as we could.*
(F, 38)


## 4. Discussion

The COVID-19 pandemic has had an impact on families globally, but particularly on the families of FHWs who experienced increased workloads, increased stress and fear regarding potential transmission of the virus to themselves and their families. Despite this, a strong sense of purpose and dedication to their work was conveyed.

Key changes in family functioning observed by participants included shifts in roles and routines within their families, some of which were made due to lockdowns and others that were made to reflect changes in their workloads and availability. Households spent more time together and often reported increased feelings of closeness as a result. Frontline status, however, sometimes interfered in closeness, with a reduction in hugs and additional washing when parents returned home from a shift to protect their families from potential transmission. Mental health and wellbeing appeared to suffer. In response, some made a conscious effort to improve coping by decreasing expectations within the household.

### 4.1. A Time of Change

Consistent with existing research, participants typically described having more time with family members within the household during the pandemic, particularly during periods of lockdown [13,34,35,36]. One potential benefit of increased time together is an improved understanding of each other’s worlds [13]. Participants reported that they felt closer to other family members and “more present” following time together. These findings underscore the importance of communication and meaningful time together where family members are physically and cognitively present. These results align with previous research reporting that family functioning and connection improved for some families during the COVID-19 pandemic due to increased communication [37,38] and time together [39,40,41].

Nonetheless, a number of studies have demonstrated that increased time together has exacerbated challenges in some vulnerable families [42,43,44,45]. Brown, et al. [46] found that parents experienced cumulative stressors through COVID-19, which impacted on their mental health and increased child abuse potential. Likewise, Evans, et al. [36] found that the COVID-19 pandemic increased parents’ stress and decreased their opportunities for respite. In this study, FHWs with vulnerabilities such as sole parenting, those with children with additional needs and families with multiple births experienced an exacerbation in their usual concerns during periods of lockdown, likely impacted by the increased isolation and decreased support available.

Time together also came at the expense of time alone, decreasing opportunities for respite and often impacting wellbeing. This seems particularly salient for healthcare workers who were leaving complex, busy workplaces and then coming home without time to debrief with peers or manage their wellbeing in their usual ways due to lockdown regulations. Zaçe et al. [47] highlight the importance of healthcare workers balancing their own self-care and wellbeing needs with the needs of others, as failure to do so can impact the individual, their family and the quality of their patient care. The importance of debriefing and other forms of self-care has been emphasized in research across health disciplines, including groups of doctors [48] and allied health staff [49,50]. Limiting time and opportunities to engage in self-care during lockdowns could therefore have significant impacts on the wellbeing and mental health of FHWs and should be carefully considered by healthcare organisations and policymakers. FHWs who are also parents may be particularly vulnerable, as attending to the needs of young families may become a barrier to attending to their own wellbeing, practically, in terms of time, but also in terms of permission to seek help.

Several initiatives to support FHWs’ wellbeing have been put into place nationally and internationally through the course of the pandemic. The Black Dog Institute in Australia, for example, provides online resources for healthcare workers in support of their own mental health [51]. Similar online psychoeducation resources have been identified internationally; see [47] for a review. For those suffering significant burnout and/or significant distress however, online resources may not be enough. Other approaches such as the provision of separate accommodation to protect family members [52], music therapy [53], online therapy [54], peer support [55,56], and mindfulness groups [57] have therefore been trialled internationally. As the mental health impacts of pandemics on FHWs are complex, multifactorial, sustainable interventions are needed to support worker wellbeing. Efforts should be increased during periods of lockdown and flexibility carefully considered for this vulnerable cohort.

### 4.2. Roles and Routines

Juggling the demands of home schooling and the subsequent impacts on workload was a salient theme for FHWs, often increasing stress. Consistent with existing research, home schooling was most impactful for parents of primary school children, who required additional supervision and support [58]. Similarly, Kallitsoglou and Topalli [59] found that home schooling was also a source of stress for working mothers in the UK. Protective factors identified in the study included having a partner, flexible work arrangements and school support. However, for the frontline healthcare workers interviewed in the present study, flexible work arrangements to meet the needs of home schooling were not possible. Additionally, the FHWs in this study reported increasing, not decreasing, workloads during periods of home schooling, adding an additional load at a challenging time.

The importance of handing over workload must be considered for parents struggling with the demands of home schooling and/or frontline work. Some participants were challenged by the need to hand over existing roles and responsibilities to other family members, even when support was offered. Others simply did not have the supports available. Encouraging healthcare workers to hand over tasks when they are busy and increasing their capacity to access outside support during lockdowns are vital to the support of this workforce. Even the provision of basic functional supports such as groceries, at-home cleaning or gardening and childcare may help these workers to juggle the demands of an increasing workload while also supporting their families. When support was not accessed, the demands of balancing multiple roles led to increased parental stress in the sample, which has also been noted in previous research [60,61,62].

The need for constant adaptation in routine was another prominent concept that permeated FHWs’ observations of their family life during the pandemic. Specifically, participants reported the need to constantly adapt routines and procedures in their workplace only to return home and have to adjust family routines. Family routines are one factor thought to play an important role in family functioning, scaffolding family interactions and protecting family wellbeing [58,63]. Bates et al. [64] found that engagement in family routines could play a protective role during the pandemic, effectively buffering families from the impacts of COVID-19-related stress. At best, the constant shifts identified may have decreased participants’ access to a valuable protective factor. At worst, the highly elevated levels of stress associated with constant change could have made the task of positive leadership in the family challenging, if not insurmountable [65].

### 4.3. Frontline Work

An important outcome of this study is an increased understanding of the impacts of the pandemic, not just on FHWs but on their families and family relationships. Many participants observed significant, sometimes detrimental impacts of their role on their family life. Participants reported increased hypervigilance at home in order to mitigate the risk of transmission to family members. This is not surprising, given that no vaccine was available to decrease the severity of illness at the time of the interviews. They also reflected on the changes in their workplace that either heightened or reduced their anxiety. At the start of the pandemic, limited access to appropriate PPE combined with uncertainty and a lack of clear information regarding COVID-19 led to increased fears of transmission. PPE access has been a common concern identified through the literature, observed across disciplines and different countries [66,67]. Positively, most reflected that their anxiety actively decreased when PPE became more accessible.

In previous studies, anxiety regarding expectations that staff practice outside their scope of practice were expressed [68]; however, they were not identified in this study. As noted earlier, Australia reported very low numbers of COVID-19 infection at the time the interviews were conducted. Reduced rates of infection relative to other parts of the world may have served a protective function in this instance, as FHWs were not practicing outside of their scope of practice as routinely. There was, however, an increase in workload, with many participants reporting a sense of obligation to pick up shifts to support the “team”. Increased workload led some parents to feeling “spent” and “less tolerant” at home. In some instances, this led to harsher parenting. Others have found an increase in “harsh” or punitive parenting during the pandemic [69,70], leading in some cases to child maltreatment and abuse [46,71]. Risk factors in these instances include job loss [72], income reduction and financial concerns [46,73,74,75] and increased alcohol consumption [70]. While frontline status has not been identified as a factor contributing to more extreme forms of maltreatment such as abuse, reports of decreased tolerance are an important indicator for early intervention.

For healthcare workers, their roles also appeared to provide an additional source of pride and respect from others. Pride was conveyed in the way in which participants spoke about their roles. While many could acknowledge the stressors associated with their work roles and many expressed fears related to their safety and that of their family, they also conveyed a sense of assurance that this experience was “part of the job”, and few reflected that the dangers would change their commitment to their healthcare roles. These findings align with a systematic review from Billings et al. [66], who reported that concurrent with workload and other pressures, healthcare staff described aspects of their jobs as rewarding, important and meaningful.

Respect in their healthcare role was reflected in family members seeking out advice and reassurance from participants during the pandemic and having a sense that they had something to contribute. In some cases, fear led to the stigmatisation of FHWs, with some extended family members creating distance due to perceived danger. Concerns regarding the mental health impacts of frontline roles on healthcare workers and the potential for avoidance of healthcare workers by family or community owing to stigma and fear were raised by WHO in March 2020 [22]. Although concerns regarding stigma were not raised universally in this sample, it is concerning that at the time of interview, more than 15 months into the pandemic, stigma was a concern for some.

### 4.4. Family Resilience and Connection

A premise of family systems theory is that serious life events and challenges, such as a pandemic, impact the whole family. Further, key family processes are thought to mediate adaptation to these events, both for individuals for and the family unit [76]. This study highlights changes in the families of FHWs that are significant and distinct from those observed in Australian family functioning studies using other cohorts [13,34]. Although some of these challenges were problematic for FHWs and their family members, such as hypervigilance and anxiety, other positive changes were noted that suggest the emergence of positive adaptation and resilience.

According to Walsh, family resilience refers to the capacity of the family to withstand and rebound from adversity [76,77,78]. Resilience involves more than coping, shouldering the burden or surviving adversity. It involves the potential for positive transformation on both an individual and a relational family level [76]. Resilience was identified within the study, though it fluctuated for many. As an illustration, lowering expectations of themselves and their parenting role appeared to help as an active coping mechanism for some FHWs, effectively reducing the pressure they might normally place on themselves, thereby providing space for resilience to develop. Resilience was also seen in family flexibility, for example, finding new ways to spend time together during lockdown and identifying ways to stay safe, such as additional washing and hygiene at home, but still stay together and connected. Communication between parents was also noted in navigating the changes and for some couples, creating positive change.

Consistent with existing research [13,36], this study again highlighted the importance of the family meal as an important ritual in family life. Frontline families reported having more time to engage in a prepared family meal during lockdowns due to a decrease in external activities. There also appeared to be an element of mindfulness in the activity, as families highlighted an awareness of its importance as a ritual of connection.

### 4.5. Gender

Two small but significant gender differences were noted in the data. First, male FHWs reported less stress and change associated with home schooling than their female counterparts did. While small changes in workload to assist their partners at home were reported, the majority of male FHWs already had systems in place that allowed for the impact of their work role on the family unit. By contrast, females in the study tended to pick up responsibility for home schooling and other changes on top of their healthcare roles.

Female participants found that they typically picked up home schooling on top of their workplace role, leading to heightened stress. In families that managed better, their partners were able to take on more when they were at home. The findings align to some extent with Nishida et al. [25], who found that the pandemic exacerbated gender inequalities. Specifically, they noted that physician mothers were caught in a dilemma between increased home duties and increased clinical hours during the pandemic, certainly more so than physician fathers.

### 4.6. Limitations

No participants had tested positive to COVID-19 at the time of the interviews; thus, they could not comment on the direct impact of COVID-19 infection on family functioning. As noted in the introduction, this finding is not surprising given the extremely low case numbers in Australia at the time of these interviews. Future studies will likely remedy this concern given the prolific spread of the BA.2 Omicron subvariant in Australia in late 2021–2022. Further, over half of the sample were female, of Australian background and in nursing roles. With respect to discipline and gender, studies suggest that the frontline healthcare workforce in hospitals skews towards nurses [79], most of whom are female [80,81]. While attempts were made to note the potential impacts of gender and discipline on responses, the stratification of the sample should be considered in reading.

Future studies need to investigate the experiences of the whole family system, taking into account the perspectives of partners and children to provide a more holistic view of family functioning. While an attempt was made to recruit partners and children to the study, one partner (also a frontline healthcare worker) and one adolescent had volunteered at the time of writing. Due to poor recruitment, their data were not included in the analysis. To try and recruit same family participants, the authors extended an invitation to participate to family members at the conclusion of participant interviews, both verbally and via email. Future studies could consider recruiting non-family member participants directly to provide different views on the impacts of healthcare work on family functioning during the pandemic. Future studies may also wish to broaden the working definition of family from this paper to include family members living outside of the household and consider the addition of quantitative measures to provide further breadth to our understanding of the dynamics described in this paper.

## 5. Conclusions

The COVID-19 pandemic and control measures such as lockdowns have significantly impacted family life. These impacts are particularly salient for FHWs who have had to navigate ever increasing workloads, stress and risk in their workplaces alongside home schooling and stress experienced by family members. FHWs have experienced fear attending work and have had to manage the potential implications of their roles for family members, including passing COVID-19 onto loved ones. Given the importance of this workforce and the unique challenges that they face, more must be done to support the mental health and wellbeing of FHWs and their families. Within families, interventions should also consider promoting positive adaptations such as flexibility, communication and connection in family units. Systemic supports are also critical and should be co-designed with FHWs and their family members to ensure optimal engagement.

## Figures and Tables

**Figure 1 ijerph-19-04897-f001:**
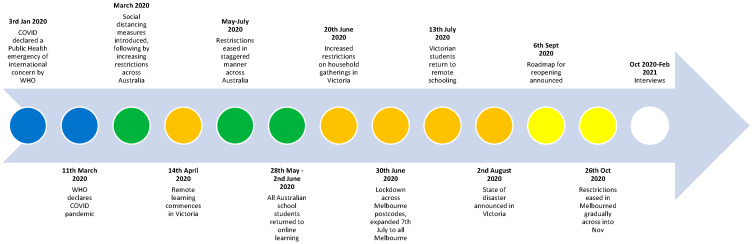
Key milestones in Australia’s response to COVID-19, indicated by green circles, and Victorian-specific restrictions, where orange indicates increased restrictions, and yellow indicates reduced restrictions.

**Table 1 ijerph-19-04897-t001:** Participant demographics.

	*n*	% of Parents
Gender		
Female	33	85%
Male	6	15%
First language		
English	38	97%
Tamil	1	3%
Age range of participants		
Under 30	1	3%
30–34	6	15%
35–39	8	21%
40–44	12	31%
45–49	8	21%
50–54	1	3%
55–59	3	8%
Developmental/school stage of children ^1^		
Pregnant	4	10%
Birth-Preschool	15	38%
Primary	22	56%
Secondary	13	33%
Post secondary	2	5%
Relationship status		
Married/de facto	36	91%
Separated/divorced	3	9%
Cultural Background (self-identified)		
Asian	5	12.8%
Australian—non Aboriginal or Torres Strait Islander	31	79.5%
British or European	2	5.1%
Other	1	2.6%
Education		
Diploma	2	5.1%
Graduate degree	12	30.8%
Postgraduate degree	25	64.1%
Location		
Regional	6	15.4%
Urban	33	84.6%
Profession		
Allied health	14	36%
Nursing	21	54%
Physician/Medical practitioner	4	10%
Department		
COVID-19 Ward	5	13.2%
Emergency Department	23	60.4%
Hospital in the Home	5	13.2%
Intensive Care Unit	5	13.2%
Leadership position	10	25.6%
Tested positive to COVID-19		
No	39	100%
Yes	0	0%

^1^ Totals greater than 100% as several parents had children in more than one age grouping.

**Table 2 ijerph-19-04897-t002:** Themes and subthemes identified.

Theme	Sub-Themes
Time together, but at a cost	Time together Impacts on relationships Loss of personal space Loss of social connections
Increased demands	
A time of change	Changes in family routines and roles Changes in parents’ personal routines Changes in responsibilities
Hypervigilance and risk of contagion	Risk consciousness Managing and mitigating risk at home
Getting through it	Wellbeing and mental health Coping

## Data Availability

The raw, de-identified data supporting the conclusions in this paper will be made available by the authors, without undue reservation.

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
