# Peer review of "“Did You Bring It Home with You?” A Qualitative Investigation of the Impacts of the COVID-19 Pandemic on Victorian Frontline Healthcare Workers and Their Families"

_ijerph, 2022, doi:10.3390/ijerph19084897_

Round 1

Reviewer 1 Report

The authors conducted a qualitative study of frontline health workers (FHW) by interviewing 39 individuals who met their inclusion criteria during the period December 2020 and February 2021. Their primary aim was to explore the impact of the COVID-19 pandemic and associated lockdowns on FHWs and their families. Thus, they restricted enrollment to those who worked in medical settings where exposure to COVID-19 patients was high, and they additionally had to be the parent of a child younger than 18 years of age. The reason for enrolling FHWs with children was to account for the effect of the FHW role during the pandemic on family functioning.

The demographics of the 39 informants are important to note. There were significant skews in enrolled participants such that 85% were female, 91% were divorced or separated, and none had tested positive for SARS-CoV-2 to date when they were interviewed. It is not clear if their recruitment process led to such a skewed demographic, but this raises questions about the generalizability of their findings.

One major question is how “family” was defined. It seems from the author’s statements that family was defined as the FHW and their minor children with no data provided on adult children, other family members in the household, partners who either lived in the household or not, and extended family who did not reside in the same home. The issue of “family functioning” is therefore unclear regarding resources available to the FHW during the pandemic.

The authors state that they used a semi structured interview but did not provide the question prompts they used. It would be more useful had they given readers the domains that were asked and whether they corresponded to the themes that were presented. It is also unclear if survey measures were used other than collection of demographic data since they implied using research questions on family functioning.

The methods section was well written and gave evidence for iterative and reflexive procedures to ensure saturation and consensus.

The results section was difficult to tackle because the quotes and explanations were often too long and not sufficiently edited for succinct points. In the case of theme #1 “Time together but at a cost” the 4 subthemes had two positive and two negative outcomes, but they seemed to be two sides of the same coin, one theme having to do with being closed in with more interaction but loss of privacy, and the other to do with improved family relationships at the cost of those outside the family. I didn’t find the use of colloquial phrases for the superordinate themes to be on point. Line 342 should read 3.3 A world in flux rather than Changes in routine and responsibilities. In general, the results section was too long and difficult to get through because of the amount of detail. It would have helped to have some quantification of thematic results to know which ones were most frequent or less frequent to be able to assess the true impacts described.

The discussion was similarly long and repetitive with the results section. In the short section on gender, the authors homed in on an important point which has to do with their skewed sample of 85% female respondents, which is that gender inequalities could not be adequately ascertained. Similar statements about characteristics of their study sample would help contextualize their findings better, such as the very high rate of divorced/separated participants (91%) which likely influenced the stress and burdens cited in their interviews.

This leads to limitations which are not well developed. A major one is the fact that the FHW opinion/point of view was taken to represent that of the “family” as a whole and while they attempted to get minor children to agree to be interviewed, they were unsuccessful. It is unclear why they did not interview other adults in the household or in the family network for alternate views instead. The reliance on purely qualitative interview data is also a limitation in that concurrent survey measures could help better describe the mental health status of the respondents. 

The Figure 1 graphic on pandemic time events is nice but not referred to in the body of the paper other than to say it gives information about the Australian pandemic milestones.

Overall, the authors are to be commended for conducting an interesting study on FHWs and their families. There are some limitations that should be better highlighted such as the fact that any information about family functioning was provided by the FHW themselves. This may bias the results in ways that should be discussed. It is unclear if survey data were also collected but not included such as reference to research studies on family functioning and assessment. It is not stated if the researchers considering getting input from other close adult family members or partners rather than their children. If the authors could address the critiques given especially major editing of the results and discussion sections, it could be re-reviewed for acceptance.

Author Response

Thank you for your feedback - Please see the attachment for our detailed response. 

Reviewer 2 Report

Dear Authors

I hope you and your families are well, thanks for this interesting paper, please let me express my tougths and reflexions about this document. I will present them by section: 

Introduction 

In general, the introduction approaches, problematizes and appropriately justifies the object of study. A demonstration is presented where the consequences that the confinement and the policies for the containment of the pandemic in the FHW have brought. In this sense, it is considered that the research questions are relevant to this problematization.

Aims

The objectives are written in the form of research questions, which is appropriate in the case of a study of a qualitative nature and of a phenomenological type.

Section 1.3 Australian COVID context

I consider this section unnecessary, what is the purpose of including it? Since I read the document and until the end of the discussion, the information contained here is not retrieved and I consider that it does not add value to the document.

Methods 

Here I have a doubt, according to the wording it is not clear if only parents (39) and also their daughters and sons (84) were interviewed, since in table 1, information from both groups is shown, however, this is not entirely clear. It is suggested that the authors clarify the number of people interviewed.

It is suggested that the authors review Table 1. Participant Demographics, in this table there are missing data that are not indicated as “missing”, for example, the distribution by age adds up to a total of 38 cases, when it should be 39. Regarding the number of children shows the distribution of 56, however, it is indicated that it should be 84 in line 137. Likewise, the distribution by department adds up to a total of 48 when it should be 39. In general, it is suggested that the authors review with caution these data and present them appropriately in order to avoid confusion in the readers.

Procedure

I consider it a success to have thought of the questions to evaluate the change and to be able to verify that the findings were not pre-existing in the study sample and, in fact, changes derived from confinement and work in the first line of care.

2.4 Reflexivity sections seem not to be connected to the overall writing. This section does not describe the procedures that were done in order to achieve and finish the research project. 

Results

The description and name of the "Themes" can be improved, without a doubt the authors have obtained valuable information, although it seems to be presented in a "casual" way, I strongly suggest that the topics can be specified in a more adequate way to properly represent the information obtained.

Likewise, it is important that the authors define the themes and sub-themes in a more complete way, as well as accompany each of the statements that support the categories with more textual citations. For example, in the first subtopic “Time togheter” only 8 quotes are referenced. It is suggested to include more examples in order to give greater solidity to the inferences and interpretations, or where appropriate, select textual citations that more clearly reflect the concepts that the authors seek to position. For example, line 225 refers to the fact that the families made a conscious effort to carry out shared activities and as an example of this the authors place two examples "guess the time that the children would travel the perimeter of a property" and "you all kinds of conversations and ideas happen.” These two examples, from my point of view, are not very powerful actions to be able to infer that those involved were really thinking about doing shared activities.

Another example that denotes the need for more evidence is the one shown in lines 228 to 232, where reference is made to the change that occurred between phase 1 and 2 of the confinement. In this case, only one citation is exposed, so there are doubts about whether there are more, whether it was a common denominator among the study sample, or a particular situation.

This same pattern is shown in the following subtopic where the authors only use 4 quotes to justify the existence of the subtopic called “impacts on relationships”. And so on.

I consider that since it is a qualitative article, it is of the utmost importance to present a series of citations that allow us to confirm the appreciations and reflections that the authors have and that systematically accompany the readers to understand the system of inferences used by the authors to reach similar conclusions, however, this document needs to present more quotes to help confirm the interpretations.

In general, the effort made by the authors to analyze the transcripts in depth is appreciated, since this already constitutes a very large effort.

Discussion

It is considered that this section is appropriately written but does not directly address the objectives of the study, for example, the impacts of the pandemic on family dynamics and even the changes that they presented in their family functioning during the pandemic are not discussed. I consider that this information is mentioned in some way in the presentation of results, however I think it is important that it be given a leading role in the discussion.

It is also important to note that the authors refer to changes in family terms, which is not accurate, since they only interviewed health workers who attend the first line of care. Therefore, the discussion and conclusions would have to be focused in that sense and with said limitations.

Author Response

(The authors gave the same response as above.)

Reviewer 3 Report

Dear authors, it was a pleasure to read this paper. The theme is of high relevance for research, practice and intervention. The study is well-grounded, and the methodology is adequately described. The authors considered the Ethical proceedings, and the data analysis was fully detailed. The results and the discussion allow answering the questions formulated to the research. The impact of covid-19 on health professionals' mental health and their families, allied with the accomplishment and balance of their work-home responsibilities, are issues that occupational health research and intervention programs must consider most. On page 121, I believe the authors wrote, "last interview completed in 2020", meant to say 2021.

Author Response

Thank you for your feedback and engagement with this paper, and for picking up our typographical error. We appreciate your interest.